# Complex crater formation by low energy impactors

**Rodrigo Tardini Paulino** [1]*, **Thiago Oblesrczuk**[2], **Julia Lencioni Aliboni**[2],
**Annibal Hetem Junior**[2], **Jeroen Schoenmaker**[2]

**1** Department of Physics, McGill University, Montréal, Québec, Canada, **2** Centro de Engenharia, Modelagem e Ciências Socias Aplicadas, Universidade Federal do ABC, Santo André, São Paulo, Brasil

* rodrigo.paulino@mail.mcgill.ca

## Abstract

We investigate the formation of complex craters in low-energy laboratory impacts using layered granular beds and a range of impactors, including solid, liquid, and granular types. Shallow granular targets change how the impact energy is dissipated, resulting in power-law scalings for the crater diameter that depart from those observed in homogeneous targets. An adaptation of the well-known Schmidt-Holsapple scaling was made to explain the impacts made from the liquid droplets. Furthermore, we show that the layered target promotes the formation of complex crater features, including flat floors and central peaks, even at low impact energies, through an essentially distinct process when compared to high energy impacts. In particular, granular impactors consistently produce ring-shaped craters, a result explained by a mechanism analogous to air entrapment in droplet impacts. This ring-like morphology was also successfully reproduced in simulations using a modelling approach developed in this work. These findings suggest that layered targets can reproduce features typical of planetary-scale complex craters at the laboratory scale, opening new avenues for small-scale experimental studies of impact dynamics with potential applications in planetary geology and civil engineering.

## Introduction

The science of impact cratering emerged in the 1960s driven by the knowledge gained from nuclear explosion tests [1]. Since then, the field has rapidly evolved through studies of explosion craters [2] and laboratory experiments involving impacts in granular materials [3]. The pivotal review by Holsapple (1993) [4] provides both a historical perspective and an introduction to the scaling laws governing impact craters formed by solid projectiles in diverse materials, including water, sand, soil (dry and wet), and rock (soft and hard).

Holsapple's work showed that the volume of a crater can be described using dimensionless parameters, the first of which being the inverse Froude number,

**Data availability statement:** Data from the experiments studied in this paper are available through Zenodo: Paulino, R. (2025). Data of complex crater diameter obtained by impact from liquid droplet, solid sphere and granular impactor [Data set]. Zenodo. https://doi.org/10.5281/zenodo.15484821.

**Funding:** The author(s) received no specific funding for this work.

**Competing interests:** The authors have declared that no competing interests exist.

$Fr^{-1} = gD/U^2$, where $D$ is the projectile diameter, $U$ its velocity and $g$ the gravitational acceleration. This parameter captures the ratio of gravitational (or lithostatic) to dynamic pressure during impact and has been widely adopted in impact studies [5,6], though it is sometimes defined with an additional factor of 2 [7–9]. In planetary craters, $Fr^{-1}$ typically ranges from $10^{-6}$ to $10^{-2}$ [4,8].

The composition and structure of the target material also plays a crucial role in determining the crater characteristics. Impacts into solid rock produce vastly different results compared to those into soft and granular targets [10,11]. Furthermore, in planetary cases, targets are most often inhomogeneous and not uniform. For example, the rather squared shape of the Barringer crater has been attributed to anisotropies caused by tear faults across the site of the crater [12]. Other studies have examined the influence of inhomogeneous or layered targets, including coarse-grained materials, stratified substrates [13,14] or attempting to replicate the surface layers of the moon [15]. These structural complexities are also central to modelling large impact basins like Chicxulub and Orientale (both estimated as $Fr^{-1} \sim 10^{-4}$ events), where layered targets are assumed to explain their multi-ring features [16,17]. These significant effects have made the analysis of craters an ubiquitous first step for determining the properties of a lunar or planetary surface. The unusual shapes of craters served as the first evidence for subsurface ice in Mars [18] and for the icy surfaces on the satellites of gaseous giants Jupiter [19], Uranus and Saturn [20]. It is also an important method of mapping the regolith on the moon [15,21–23].

Craters with unusual morphologies are usually associated with low $Fr^{-1}$ and high-energy impacts. These so-called complex craters present structures with flat floors, central peaks, scalloped rims, or terraced walls, that deviate significantly from the simple bowl-like shape [24]. The formation of these features has been qualitatively compared to what happens when a droplet hits a fluid surface and a tall central peak is formed, which becomes a source to surface oscillations that drive these complex features [25–27]. Because of their intricacy, these morphologies are often studies through numerical simulations rather than analytical models [5,23].

Despite their smaller scales, laboratory experiments have successfully reproduced many features of large-scale impacts, validating scaling relations and providing necessary information for the development of more accurate models [28–30]. These studies have investigated a wide range of variables including impact velocity [6,31], projectile size and density [32], shape [33], and material type-including liquid and granular projectiles [8,28]. Yet, systematic studies of the target material have been limited to changing the particle size in granular beds [7] or the porosity of sedimentary rocks [29], while the role of target inhomogeneities was only explored in moderate energies and very low inverse Froude numbers ($Fr^{-1} \sim 10^{-8}$) [15].

In this work, we investigate shallow granular beds as a naturally relatable form of layered and inhomogeneous targets. We focus on low-energy impacts ($10^{-2} < Fr^{-1} < 10^0$) into beds composed of a granular surface layer above a solid, impenetrable substrate (as seen in Fig 1a). We find that when the bed depth is less than the impactor's penetration depth, energy dissipation is altered and the well-established Schmidt-Holsapple scaling must be modified to accurately predict crater diameters. Due to the similarities in the experimental setup, we briefly discuss

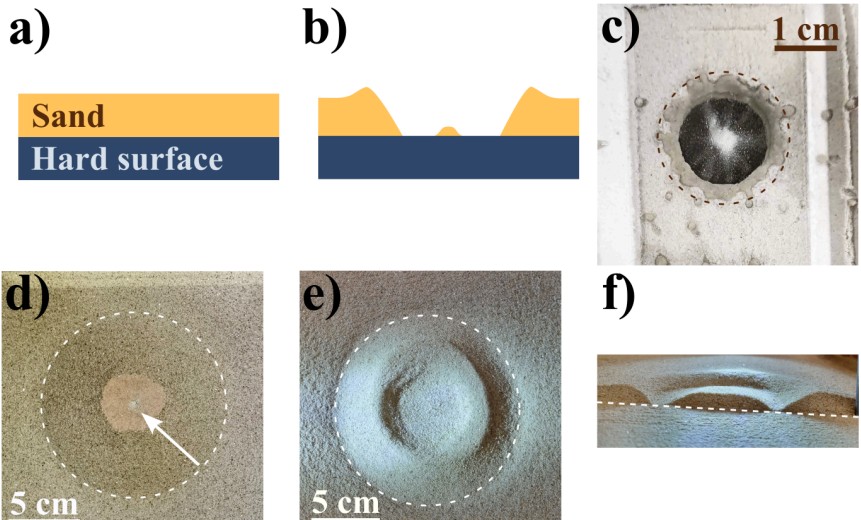

**Fig 1**. **Experimental layout and typical craters.** Side view sketch of a layered inhomogeneous bed comprised of a granular surface and a high strength, essentially impenetrable, subsurface, rendering a shallow granular bed a) before and b) after impact where we observe a complex crater with flat floor and a central peak. Photograph of craters obtained with impacts from c) a water droplet with 0.36 mJ of energy, d) a solid steel sphere with 8.6 J of energy and e) a granular impactor with 9.7 J. In c) and d), a central peak can be seen (as displayed by the white arrow), and in e), a ring-shaped crater is obtained. See Methods section for details and Results for the discussion on the morphology. f) Cross section of crater obtained with the granular impactor, where the annular shape becomes apparent.

the implications of this effect in the field of civil engineering, where a sand cushion layer is commonly used to reduce the effects of rockfall in concrete slabs.

Additionally, by using solid, liquid, and granular impactors, we demonstrate how stratified beds affect crater formation and lead to complex morphologies, such as ring-shaped craters with flat floors and central peaks (see Fig 1). While these features are typically associated with high-energy impacts, we show that they can also arise in low-energy regimes. We complement our experimental results with numerical simulations using genetic algorithms to explore the parameter space and support our findings.

## Methods

### Experimental methodology

We carried out experiments using liquid, solid and granular impactors consisting of water droplets, a 3.00 cm diameter steel sphere weighing 110.07 g, and a latex balloon filled with sand (from now on referred to simply as the granular impactor) weighing 110 g, rounded up in the shape of a sphere prior to each launch.

**Liquid impacts.** The experiments with liquid impactors were done by fixing a dropper that consistently delivered water droplets of diameter $D = 5.6 \pm 1.0$ mm at different heights. The dropper height can be directly related to kinetic energy by measuring the final speed of the droplet right before hitting the granular bed, which was done by tracking film footage obtained with a digital camera at 960 fps. This method was preferred over the direct conversion of potential to kinetic energy in order to consider the effects of air resistance. The granular bed is composed of glass microspheres, with average diameter $d_s = 53$ $\mu$m, filling ratio $\phi = 0.60$ and density $\rho_s = 2.5$ g/cm$^3$. The thickness of the granular layer was ensured by using masks of known thickness, and these masks were done by stacking bond paper sheets with rectangular holes cut on them. The granular bed was prepared by spreading the glass beads on a flat mirror, and by levelling the beads to the mask. The crater diameters were determined by fitting a circle to scaled images of the crater, disregarding the shape

and size of the granular residue in the center, similarly to previously reported studies [6,28,34]. Six different impact experiments were performed for each data point to average over bed inhomogeneities and obtain error estimates. An example of such impact is shown in Fig 1c.

**Solid and granular impacts.** The experiments with the steel sphere and the granular impactor were performed by dropping the projectile from a controlled height and the impact energy was calculated using the equation $E = mgh$, where $E$ is the energy, $m$ is the mass of the impactor, $g$ is gravity's acceleration and $h$ is the dropping height. For the bed, we used a $55 \times 55 \times 12$ cm$^3$ wooden box filled with sand with the desired depth. The sand was sifted through a 1 mm mesh and dried up on a hotplate set to 200 ℃ for 2 hours. Prior to each launch, the sand was stirred up throughout the entire box and then gently levelled for compactness consistency. Similarly to the case of liquid droplets, we performed six impact experiments for each condition for the sake of statistics, and crater diameters were determined by fitting a circle to the highest points of the rim. An example of such impact with the steel sphere is shown in Fig 1d, and with the granular impactor in Fig 1e and 1f. In order to obtain cross section of the ring-shaped craters, half of the crater structure was knife-edged away in order to reveal the characteristic side view profile.

### Numerical simulations - Granular impact

With the goal of understanding the formation of the complex craters from a granular impactor, we have performed computational simulations of a spherical quantity of sand impacting a surface of the same material. Usually, in the crater formation literature, the high energies involved make all granular matter behave essentially as a liquid, allowing computational fluid dynamics algorithms to be implemented. This approach would not be valid in the significantly lower energy regimes employed here, so we opted to develop our own custom framework to simulate the crater formation process. The simulation procedure is thoroughly described in the Supporting Information, but a brief overview is given here.

The primary objective was to determine how input parameters — including the falling velocity of the sand, the friction between grains, and their elastic coefficient — influence the geometry of the crater formed after the impact. The adopted approach enabled the generation of graphs correlating the input variables with the resulting characteristics, providing a detailed understanding of the physical interactions involved in the process.

For such, we have used a physically accurate particle tracking software (Howling Moon Software [35]), recording properties of the individual sand grains, such as position and rotation, during the interactions with neighbouring grains. Standard tests, as the dilatancy model test [36], were also performed for adjustment and calibration, ensuring that the outcomes accurately reflected the physical conditions of the model.

A series of simulations were performed to model the impact of a sand sphere against a sand-covered surface. For each simulation, a comprehensive analysis was conducted in the following steps: (1) identification of the surface as the grains with the smallest number of neighbours; (2) based on this set, an interpolation function was generated with third-order splines, eliminating discretization from the nature of the grains; (3) the resulting function was then fitted to the complex crater model to obtain the geometric parameters according to the model. The extracted parameters of the complex crater were the height of the central peak, the radius and depth of the crater, and the radius and depth of the depression, which is the bottommost region of an annular crater.

The large parameter space and the variety of results it produced allowed the development of a genetic algorithm code to search for the initial parameters that more accurately reproduce our experimental results [37]. Details of the genetic algorithm and the obtained input parameters are again provided in the Supplementary Information. The genetic algorithm procedure reached the optimized parameters of 13.25(4) m/s for the impact velocity, 0.18(2) for the unit-less grain elasticity and 0.98(5) for the unit-less grain friction.

After optimization, the model demonstrated a robust capacity to reproduce the expected behaviour within the defined constraints, validating the reliability of the optimization process and highlighting the potential for accurate predictions of

the scenarios of interest and possibly even further applications involving granular materials in engineering, geophysics and planetary science.

## Model - Adapted Schmidt-Holsapple scalling

In this section, we describe the model used to explain the impact of liquid droplets into inhomogeneous beds consisting of a shallow granular layer on top of an impenetrable one. Based on quantitative similarities of impact craters formed by liquid droplets to high-velocity asteroid strikes, Zhao *et al.* have employed the well-known Schmidt-Holsapple (S-H) scaling of high-energy planetary impacts to the case of simple craters in homogeneous granular beds [2,28]. The S-H scaling can be derived by making three main assumptions: (i) a small fraction of the total kinetic energy is used to eject the particles ($E_{ej} = fE$), while most of the energy is dissipated in secondary processes proportional to the area of the crater. These range from deforming the droplet surface energy to viscous dissipation, and results in a phenomenological definition of $f = \pi D^2/\pi D_c^2$, where D is the droplet diameter and $D_c$ is the crater diameter; (ii) the energy required to form the crater is determined only by the mass of the particles that formed the bed before impact and the volume of the crater; (iii) the crater can be approximated to a paraboloid with an experimentally determined aspect ratio between the depth of the crater $d_c$ and the outer diameter $D_c$ as $d_c = \alpha D_c$. With these considerations, Zhao et al. showed that the Schmidt-Holsapple scaling for homogeneous granular beds is:

$$D_c \approx \left(\frac{\pi}{8}\alpha^2\phi\rho_s g\right)^{1/6} D^{1/3}E^{1/6},$$ (1)

where $D$ is the diameter of the droplet, $E$ is the kinetic energy before impact, $\alpha$ is a constant determined experimentally as $\alpha \approx 0.2$ for both liquid droplets and asteroid impacts [28], and the filling ratio $\phi$ and the density of the granules $\rho_s$ are fixed experimental parameters determined by the material that composes the bed. Complex craters with flat floors present different values for $\alpha$ [24], but we argue that it is still a valid approximation considering that we correct the paraboloid shape to one with a flat floor, as will be described below.

In the scenario where we have a shallow granular bed of thickness $t$ on top of an effectively impenetrable hard surface, assumptions (i) and (iii) must be adapted. On assumption (i), one now has to consider that an additional part of the energy will be lost to the adhesion of water to the surface. In practice, this means that a thinner granular bed will render a lossier impact process, so we expect the fraction of energy $f$ used to eject particles to increase with the thickness $t$. We propose

$$f = \begin{cases} \beta\frac{D^2}{D_c^2}\frac{t}{\alpha D_c}, & \text{if } t < \alpha D_c \\ \beta\frac{D^2}{D_c^2}, & \text{else} \end{cases}$$ (2)

where $\beta$ is a phenomenological fitting parameter of the model that encompasses the additional energy dissipation.

Assumption (iii), on the other hand, can be corrected by approximating the shape of the complex crater as a cross section of a paraboloid where the height is between $t$ and $\alpha D_c$. The resulting volume can be calculated as $V = \pi t D_c(D_c - t/2\alpha)/4$, where we end with the following polynomial equations:

$$\begin{cases} D_c^5 - \frac{t}{2\alpha}D_c^4 - \frac{C\beta D^2 E}{t\alpha} = 0, & \text{if } t < \alpha D_c \\ D_c^4 - \frac{t}{2\alpha}D_c^3 - \frac{C\beta D^2 E}{t^2} = 0, & \text{else} \end{cases}$$ (3)

where $C = 4/\pi\phi\rho_s g$. The roots of these equations can be calculated numerically and used to fit our experiments.

This model, however, does not account for what happens at very small thicknesses, as the energy of the surface deformation and adhesion of the droplet to the hard surface becomes much more significant than the energy required to create the crater. In that case, the splash of the droplet coupled to the adhesion to surface is what will determine the maximum

diameter [38,39]. It is, however, a valid approximation for regimes of intermediate thickness and provides an phenomenological model that describes this specific type of complex crater, as will be shown through the experimental results.

## Results and discussion

### Crater diameter

**Liquid impacts.** Fig 2a displays the crater diameter as a function of sand layer thickness, obtained with three different liquid droplet energies. An overall trend can be seen of increasing diameter with shallower granular beds (or smaller sand thicknesses), where we obtain complex craters with flat floors. This is contrary to the regular Schmidt-Holsapple scaling, that doesn't predict a dependency with bed thickness [2,28], but can be expected considering that the energy injected by the droplet into the granular bed has a smaller thickness of sand to be absorbed, so it spreads laterally making a bigger diameter crater. This is described theoretically by the adapted Schmidt-Holsapple scaling (Eq 3), as can be seen by the solid lines in Fig 2a. Considering the established connection between liquid-drop impact craters to very-high energy asteroid impacts [28], we believe this result might be of importance for the astronomical community wanting to infer planetary surface properties based on large craters, similarly to how very small craters in the moon were used to estimate the regolith thickness on the moon [15,21–23].

A departure between theory and the experimental points can be seen to the left of the vertical dashed lines in Fig 2a, representing craters obtained with very small sand thicknesses and bigger energies. This is due to a transition into a regime where the droplet adhesion to the hard surface start to dominate over the energy required to displace the material forming the crater, limiting the maximum crater diameter to the splash diameter of the liquid droplet [38,39]. Incidentally, at higher droplet energies, this transition happens at higher sand thicknesses, going from 0.4 mm and 0.6 mm for droplets with 0.21 mJ and 0.55 mJ of energy, respectively, up to 1.2 mm for 0.82 mJ droplets, resulting in an increased reduced-$\chi^2$ ($\chi^2_\nu$) for the higher energy droplets. This can also be related to the case of complex planetary craters originated by asteroid impacts, where complex craters start displaying additional features as the impact energy increases, going from flat floors to central peaks and scalloped rims [24]. In these regimes, the fluid-like properties of the droplets must be considered [5], and the adapted Schmidt-Holsapple scaling is no longer valid.

Fig 2b shows the crater diameter as a function of the droplet energy with constant sand bed thicknesses of 0.76 mm, 1.55 mm and 2.53 mm. In this scenario of constant thickness, the model falls back to the original SH scaling with a power

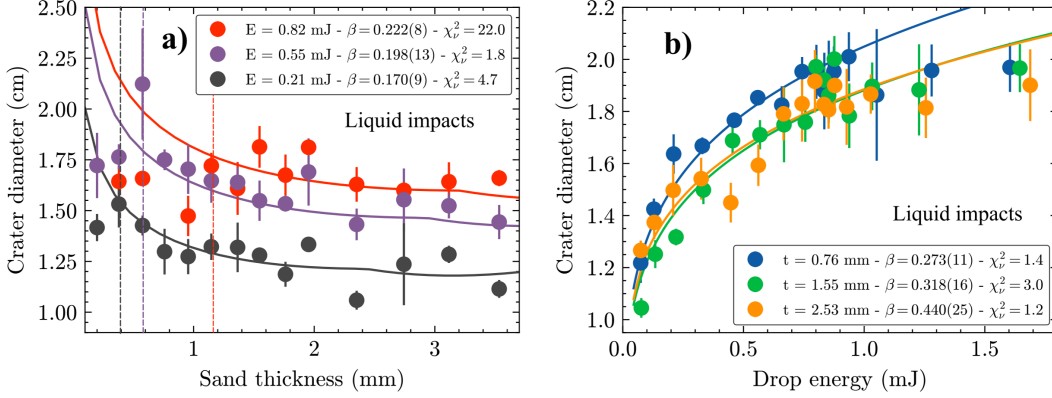

**Fig 2. Diameter of craters created by liquid droplet impact.** Crater diameter as a function of a) sand layer thickness and b) droplet kinetic energy for liquid droplets. Solid lines are fits to the adapted scaling relationship (roots of Eq 3), where the free parameter $\beta$ and the reduced-$\chi^2$ ($\chi^2_\nu$) obtained are shown on the legend. Vertical dashed lines in a) show the departure of the experimental data points from the phenomenological model due to the more significant adhesion of the droplet to the hard surface.

law behaviour of $D_c \propto E^{1/6}$, as can be more clearly seen plotted on a log scale (Fig 3b). The different bed thicknesses introduce an offset, where the shallower beds have an increase in the crater diameter, following the trend seen in Fig 2a and the adapted model.

**Solid and granular impacts.**  Fig 3a displays the crater diameters given by a solid steel sphere and the granular impactor, that represents an intermediate regime between the solid and the liquid droplet. A power law can be used to fit the experimental data, where we can observe the expected $D_c \propto E^{1/4}$ for the solid sphere in the thicker layer of sand ($t = 50$ mm, purple triangles) [4,7,31]. This value arises when an impact of energy $E$ lifts granular particles of volume $D_c^3$ to a height proportional to $D_c$. For the granular impactor (light grey squares), we can notice an overall increase in amplitude and a slight reduction on the exponent to $D_c \propto E^{0.23(2)}$, signalling that although the spreading of the impactor on impact can allow for bigger craters, a different power-law is seen because part of the energy is lost in the deformation of the surface of the balloon. Fig 3b shows the crater diameter for all different impactors in a log scale, where the different power laws are more evident.

With a smaller layer of sand ($t = 15$ mm), however, the power-law exponent is significantly reduced to a value closer to $D_c \propto E^{1/5}$ for both the sphere (red triangles) and the granular impactor (dark grey squares), showing reduced energy transfer in thinner beds. This is contrary to the behaviour shown in Fig 2 for the liquid droplet, where thinner beds had bigger diameter craters. This can be understood considering the mechanism at which craters are formed from a solid projectile, where particles absorb the impact energy and proceed to eject neighbouring particles, generating a crater that is much bigger than the projectile diameter. In a shallower granular bed, there are less particles to absorb the impact energy and more energy is lost through other mechanisms such as heating and sound. As it will be discussed in the next section, this results in craters with flat floors instead of the simple bowl-like shape.

This reduced energy transfer is of particular significance in the field of civil engineering, where a layer of sand is commonly applied on top of concrete slabs to prevent rockfall damage [40,41]. By replicating our experiments in a controlled form, an engineer might look for the presence of flat-floored craters or even analyze the scaling relation between the resulting crater diameters and the rockfall energy, where it would become apparent that the sand layer is insufficient at dissipating the collision energy. The current standard testing approach requires observing irreversible fractures in the concrete slabs [40].

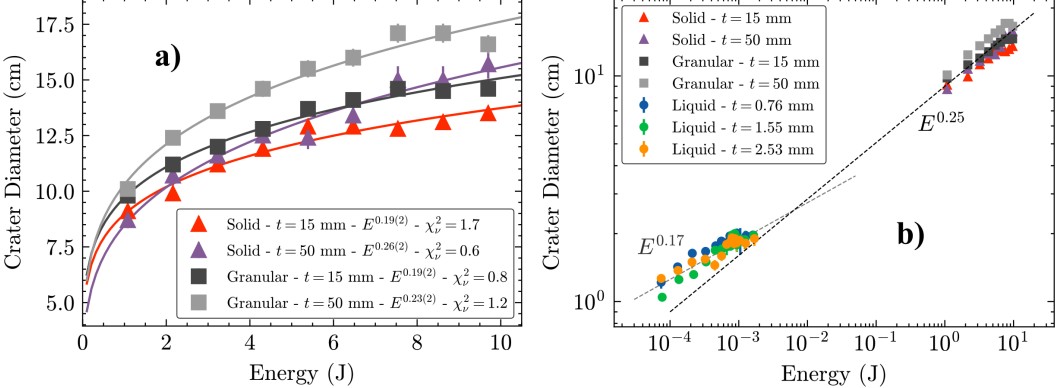

**Fig 3**. **Power law dependence of crater diameter.** a) Crater diameter as a function of energy for the solid steel sphere and the granular impactor. Solid lines are power law fittings, where the fitting parameter and the reduced-$\chi^2$ ($\chi_\nu^2$) are shown on the caption. b) Crater diameter as a function of kinetic energy in a logarithmic plot. Circles, triangles and squares represent the experiments performed with liquid droplets, solid spheres and granular impactors, respectively. Dashed lines represent the different power law relations scaling the crater formation.

## Crater morphology

**Granular impacts.** Surprisingly, our experiments with the granular impactor consistently yielded very distinct ring-shaped craters (see Fig 1e and 1f). In order to understand the phenomenon, we recurred to the physics of droplet impacts. Interestingly, we can recognize a related effect happening in the case of solid impactors as well. However, before discussing the ring-shaped crater formation mechanism, it is important to stress that high energy impact craters may be complex and present a central peak or even ringed structures (see Fig 1 of Pike [24]), but the formation process is distinct [5]. To our knowledge this is the first consistent demonstration of the formation of ring-shaped crater by the mechanism described here.

Usually, when a droplet hits a flat surface, solid or liquid, it entraps a small bubble of air underneath. Many important phenomena in droplet interactions are explained by this effect, from droplet bouncing to gas exchange between atmosphere and oceans [42,43]. When the right condition of the viscosity of the fluid and velocity of the droplet is met, the volume of air entrapped can be quite significant [44,45].

The phenomenon is explained by taking into account the axisymmetric nature of the falling process and the viscosity of the liquid drop and the air [42]. While the droplet approaches the surface, a cushion of air is formed between the droplet and the surface. As the pressure rises, there is a stagnation point in the lubricating cushion of air that deforms the drop forming a dimple in the bottom central part of the drop (Fig 4a). As a result, the impact of the droplet on the surface entraps a small bubble of air. In the case of a liquid droplet falling on a shallow layer of sand, this would sometimes result in the formation of a central peak, as shown in Fig 1c.

It is a known fact that, depending on the parameters of the process they are subjected, granular materials may present the behaviour of rigid bodies or of fluids [46,47]. In the case of the impact of granular impactors on granular beds, the materials may present a behavior that is rather akin to the one just described by the droplet and air. (Fig 4b–4f). As the granular impactor hits the sand, the bed material starts to be expelled to form a crater as normally happens in this kind of impact. However, due to the increased pressure at its bottom, the granular impactor deforms in such a way to entrap a pocket of sand underneath, similarly to the bubble of air underneath the water droplet. The main difference is that, while in the case of the droplet, the entrapped air bubble volume is usually two orders of magnitude smaller than the volume of the droplet [44,45], in the case of the granular impactor, the entrapped pocket volume is comparable to the volume of the impactor. With the sand effectively behaving as a fluid, this might be explained in terms of the differences of effective densities and viscosities involved.

Fig 1e depicts a photograph of a typical result with the granular impactor hitting a 0.5 cm layer of sand, where the impactor was carefully removed, revealing the ringed structure formed underneath. In Fig 1f the sand of half of the crater structure has been carefully knife-edged away so that the profile could be revealed and photographed. For the granular impactor, ring shaped craters have been formed consistently for impact energies above 5 J. Below that threshold, craters can more adequately be described as flat floor craters, independent of the sand bed layer thickness.

In Fig 5a, we compare the evolution of the ring crater morphology as a function of the sand bed thickness for 9.7 J energy impacts. The profiles were obtained by knife-edging the formed craters and by scale-controlled photography. As we observe for the profiles obtained for 0.5 and 1 cm depth of sand, shallower beds tend to pronounce the pocket entrapment effect, with the entrapped pocket having up to 27 % of the impactor's size. This reinforces the hypothesis that layered beds tend to induce complex crater formations, even for low energy impactors. In the case of higher sand bed thickness, although the characteristic ring-shaped crater is clearly noticeable, the overall shape tends more to a flat floor crater. This in turn, indicates that granular impactors render complex crater regardless of the bed thickness. This is consistent with experimental results reported by Pacheco-Vázquez and Ruiz-Suarez, where complex craters where reported for granular projectiles on granular beds [8]. The main difference between our results and the ones from Pacheco-Vázquez and Ruiz-Suarez, is that the central peaks and flat floor in their experiments were formed mainly from

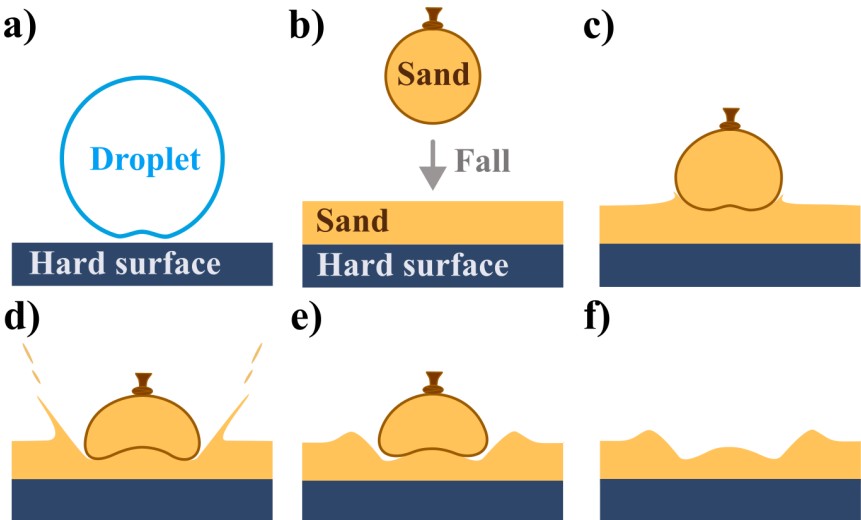

**Fig 4**. **Diagram of central peak formation.** Sketch depicting a vertical cross section (side view) of the process of pocket fluid entrapment by liquid and granular impactors. a) A liquid droplet falls onto a flat surface entrapping a bubble of air (not to scale as the volume of the entrapped bubble is usually 2 orders of magnitude smaller than the volume of the droplet). b) A granular impactor comprised of a balloon filled with sand falls onto a hard surface covered by a rather thin layer of sand. c) With the impact, the pressure of the granular bed deforms the impactor bottom-up in a dynamic manner. During the process, material from the bed is expelled outwards forming the crater. d) As the process follows on, the deformation of the impactor entraps a pocket of sand. e) The system relaxes to a steady state and the granular impactor recedes slightly. f) The resulting side view profile of a ring-shaped crater.

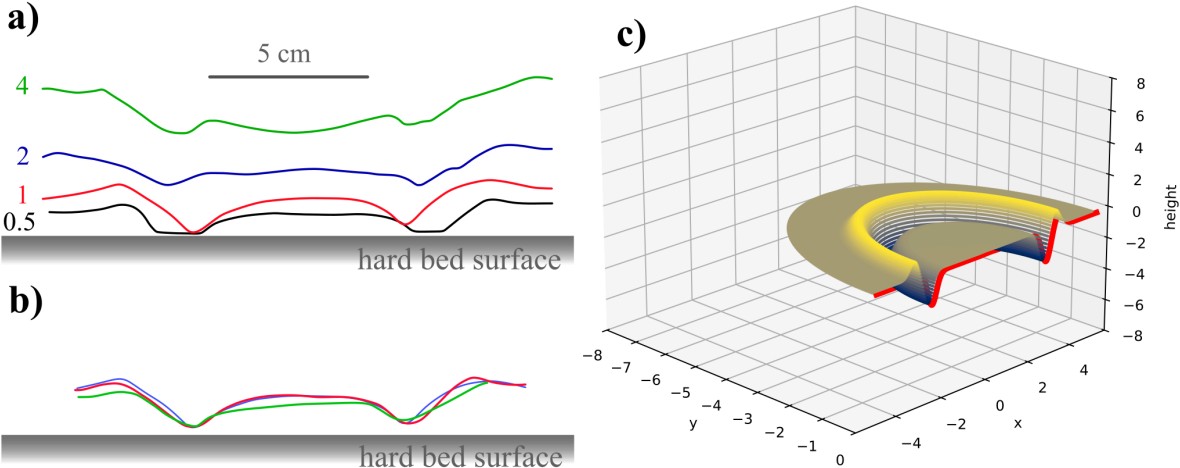

**Fig 5**. **Profile of ring-shaped crater.** Profiles of the ring-shaped craters formed by the granular impactor trials with 9.7 J of energy. a) Typical profiles obtained for different sand layer thickness namely 0.5, 1, 2 and 4 cm. The sand pocket entrapment is more effective for smaller thickness of sand layer although the ring-shaped crater is still very distinctive for 4 cm layer of sand. b) The morphology of the ring-shaped craters is consistent and reproducible as the figure shows the superposition of the profiles of three different impacts on 1 cm layer of sand. c) 3D crater profile obtained through the simulation of a granular impactor in a granular surface, as described in the methods section, showing the ring shaped morphology.

remains of the impactor, and not from the material of the bed. Fig 5b shows how reproducible the ring shape crater morphology is by showing the remarkable overlap of the profiles obtained by 3 distinct experiments in the same condition: 1 cm sand bed for 9.7 J impact energy.

Fig 5c displays the result of the optimized simulation for a granular projectile hitting a granular surface, showing the same ring shaped crater observed in the experiments. Along with the reproducibility shown in Fig 5b, the capacity of the simulation to present the same morphology suggests that this ring-like crater is a common feature of craters generated by granular impactors.

**Solid impacts.** At last, we turn our attention to the craters generated by solid steel sphere impacts, where we interestingly observed the formation of a central peak underneath the projectile, similarly to the case of the granular impactor. Fig 1d shows the crater formed by a 8.62 J steel sphere energy impact. A small pocket of sand can be seen in the middle of the flat floor of the complex crater formed (indicated by the arrow). This phenomenon was noticeable in every impact crater in which the flat floor (the hard surface) was exposed, i.e., for impact energy of 3.23 J or higher. Note that the entrapped sand pocket always shows signs of grain fracture, indicating that pressures during impact reached up to 2.1 GPa [48].

It is helpful to contextualize our experiments with earlier experiments performed by Oberbeck and Quaide, where solid spheres were impacted into a layer of granular material on top of a more cohesive but penetrable substrate, with significantly higher energies and inverse Froude numbers on the order of $10^{-8}$ [15]. Their experiments show central peaks only when the crater diameter to layer thickness ratio is around 6.3, whereas we have observed central peaks whenever the ratio is bigger than 7. We propose an explanation relating back to the entrapped bubble of air underneath the falling drop of water. If we consider the sand as a fluid during the impact process associated with the axisymmetric nature of the phenomenon, as the pressure increases, there is a stagnation point where the sand no longer can escape the very bottom of the steel ball and ends up entrapped, resulting in a next step of the energy dissipation where the portion of the remaining grains is crushed. In Oberbeck and Quaide's experiments, the penetrability of the substrate underneath allowed the pressure to be released, negating the formation of a central peak.

Focusing now on the overall structure of the crater, we observed simple bowl-like craters for all solid impacts with a $t = 5$ cm sand bed, while flat-floors were seen in the thinner sand bed ($t = 1.5$ cm). This can be explained considering that, for the thicker bed, the granular material is sufficient at dissipating all the energy of the impact without interacting with the underlying impenetrable surface, as can be observed in the crater diameter scaling with energy (see Fig 3 and discussion in previous section). This extends the observations made by Oberbeck and Quaide down to much lower energy regimes, where craters were shown to transition from a simple morphology to a flat-floored one when the crater diameter to bed thickness ratio is bigger than 4.25 [15]. For comparison, this threshold value translates to flat-floored craters being formed in craters with diameters bigger than 6.4 cm for the thinner beds or 21.3 cm for the thicker ones.

## Conclusion

We have shown that complex crater morphologies, typically associated with high-energy planetary impacts, can be reliably reproduced in low-energy, laboratory-scale experiments by using layered granular targets. Through impacts of liquid droplets, solid spheres, and deformable granular projectiles, we observed that crater diameter scaling depends critically on both impactor type and bed thickness. Liquid droplets followed an adapted Schmidt-Holsapple scaling, with deviations at thin bed layers where adhesion effects dominate. Solid and granular impactors revealed a shift to lower scaling exponents in thinner beds, highlighting reduced energy transfer efficiency compared to liquids, in a similar fashion to what was observed on higher energy experiments.

Most notably, we consistently observed the formation of ring-shaped craters from deformable granular impactors, attributed to the entrapment of sand pockets during impact, analogous to the mechanism of air bubble entrapment in droplet impacts. Numerical simulations confirmed the robustness of this effect. We also found evidence of similar sand entrapment under solid spheres, a phenomena that persisted in a much bigger energy range than the ones previously observed in low inverse Froude number experiments.

This work demonstrates that layering in granular beds is a simple yet powerful method to induce complex crater features at a very wide range of energy scales, opening new pathways for controlled experimental studies of impact dynamics and crater morphology, that are of interest to the fields of planetary geology and civil engineering.

## Supporting information

**S1 File. Supporting information.** Modelling of granular projectile impact.
(PDF)

## Acknowledgments

We gratefully acknowledge the International Physicists' Tournament (IPT) for proposing the original problem that inspired this research. We also thank the technicians at UFABC's Physics Laboratory, Heriques Frandini, Pedro da Silva, Daniel de Aquino and Eduardo Campos for their invaluable support in setting up the experiments. Finally, we extend our appreciation to the members of the UFABC IPT team for their enthusiasm, collaboration, and critical discussions throughout the development of this work. The APC of this manuscript was sponsored by Terra Viva.

## Author contributions

**Conceptualization:** Rodrigo Tardini Paulino, Thiago Oblesrczuk, Jeroen Schoenmaker.

**Data curation:** Rodrigo Tardini Paulino, Thiago Oblesrczuk, Julia Lencioni Aliboni, Jeroen Schoenmaker.

**Formal analysis:** Rodrigo Tardini Paulino, Annibal Hetem Junior, Jeroen Schoenmaker.

**Investigation:** Rodrigo Tardini Paulino, Thiago Oblesrczuk, Julia Lencioni Aliboni, Annibal Hetem Junior, Jeroen Schoenmaker.

**Methodology:** Rodrigo Tardini Paulino, Annibal Hetem Junior, Jeroen Schoenmaker.

**Project administration:** Jeroen Schoenmaker.

**Resources:** Rodrigo Tardini Paulino, Jeroen Schoenmaker.

**Software:** Annibal Hetem Junior.

**Supervision:** Jeroen Schoenmaker.

**Validation:** Rodrigo Tardini Paulino, Jeroen Schoenmaker.

**Visualization:** Rodrigo Tardini Paulino, Annibal Hetem Junior, Jeroen Schoenmaker.

**Writing – original draft:** Rodrigo Tardini Paulino, Annibal Hetem Junior, Jeroen Schoenmaker.

**Writing – review & editing:** Rodrigo Tardini Paulino, Jeroen Schoenmaker.

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
