## [Decision Letter · Decision Letter 0]

2 Jul 2025

PONE-D-25-29855Complex crater formation by low energy impactorsPLOS ONE

Dear Dr. Tardini Paulino,

Thank you for submitting your manuscript to PLOS ONE. After careful consideration, we feel that it has merit but does not fully meet PLOS ONE’s publication criteria as it currently stands. Therefore, we invite you to submit a revised version of the manuscript that addresses the points raised during the review process.

We look forward to receiving your revised manuscript.

Kind regards,

Guojin Qin

Academic Editor

PLOS ONE

Journal Requirements:

Reviewers' comments:

Reviewer's Responses to Questions

**Comments to the Author**

1. Is the manuscript technically sound, and do the data support the conclusions?

Reviewer #1: Partly

Reviewer #2: Yes

2. Has the statistical analysis been performed appropriately and rigorously?

Reviewer #1: No

Reviewer #2: Yes

3. Have the authors made all data underlying the findings in their manuscript fully available?

Reviewer #1: Yes

Reviewer #2: Yes

4. Is the manuscript presented in an intelligible fashion and written in standard English?

Reviewer #1: Yes

Reviewer #2: Yes

5. Review Comments to the Author

Reviewer #1: Reviewed manuscript presents a study of crater formation in granular beds under the impact of a low-energy impactor. The experimental results are complemented by numerical simulations using genetic algorithms.

The experiment presented is correctly and interestingly designed. The results obtained appear to be reliable. However, the paper lacks justification for the research undertaken. The authors have not explained the practical aspect of the proposed research. This is, in my opinion, a significant shortcoming of this manuscript.

In addition, the literature review performed is not sufficient. The authors have not included an adequate reference to the work of previous researchers, either to outline the background to their own research or to support the results obtained.

The most significant major shortcoming of this article is the discussion, which is conducted as a description of the results obtained rather than a polemic against the results of other researchers. I consider this to be a significant shortcoming that prevents the article from being published in its current form.

I suggest completing the literature review, justifying the undertaking of the present research topic and thoroughly rewriting the text, especially in the Discussion and Conclusion sections.

Reviewer #2: Dear authors

Any experiment is worth checking and publishing. Hence, I am suggesting only a minor revision. In this regard what you have to do while resubmitting is that you have show real examples for your experiment. If you are not getting examples from Earth, you can choose Moon or Mars.

Wishes

6. PLOS authors have the option to publish the peer review history of their article (what does this mean?). If published, this will include your full peer review and any attached files.

Reviewer #1: No

Reviewer #2: No

---

## [Author Response · Author response to Decision Letter 1]

28 Jul 2025

We address the reviewers' comments individually in the "ResponseLetter.pdf" file.

---

## [Decision Letter · Decision Letter 1]

9 Sep 2025

PONE-D-25-29855R1Complex crater formation by low energy impactorsPLOS ONE

Dear Dr. Paulino,

Thank you for submitting your manuscript to PLOS ONE. After careful consideration, we feel that it has merit but does not fully meet PLOS ONE’s publication criteria as it currently stands. Therefore, we invite you to submit a revised version of the manuscript that addresses the points raised during the review process.

We look forward to receiving your revised manuscript.

Kind regards,

Dr. Guojin Qin

Academic Editor

PLOS ONE

Journal Requirements:

Reviewers' comments:

Reviewer's Responses to Questions

**Comments to the Author**

1. If the authors have adequately addressed your comments raised in a previous round of review and you feel that this manuscript is now acceptable for publication, you may indicate that here to bypass the “Comments to the Author” section, enter your conflict of interest statement in the “Confidential to Editor” section, and submit your "Accept" recommendation.

Reviewer #1: (No Response)

2. Is the manuscript technically sound, and do the data support the conclusions?

Reviewer #1: Partly

3. Has the statistical analysis been performed appropriately and rigorously?

Reviewer #1: No

4. Have the authors made all data underlying the findings in their manuscript fully available?

Reviewer #1: Yes

5. Is the manuscript presented in an intelligible fashion and written in standard English?

Reviewer #1: Yes

6. Review Comments to the Author

Reviewer #1: I would like to thank the authors for responding to my previous comments. The changes made have improved the legibility of the manuscript.

Due to the diversity of the experiments conducted and numerical simulations, the text is still difficult to fully understand in places. I therefore suggest introducing a division based on the impactor used: i) liquid, ii) solid, iii) granular, both in the description of the methodology and in the Results and Discussion.

In my opinion, this minor change in the text and figures will significantly improve the readability of the article.

Please remove from the text your own descriptions (well understood by the author, but difficult for the reader to understand unambiguously), such as: described as an inverted rim placed inside the outer rim (256).

Please also add a statistical analysis of the results obtained.

7. PLOS authors have the option to publish the peer review history of their article (what does this mean?). If published, this will include your full peer review and any attached files.

Reviewer #1: No

---

## [Author Response · Author response to Decision Letter 2]

16 Oct 2025

The response letter to the reviewer is in the file ResponseLetter.pdf. All comments are addressed there.

---

## [Editor Report · Decision Letter 2]

20 Oct 2025

Complex crater formation by low energy impactors

PONE-D-25-29855R2

Dear Dr. Paulino,

We’re pleased to inform you that your manuscript has been judged scientifically suitable for publication and will be formally accepted for publication once it meets all outstanding technical requirements.

Kind regards,

Dr. Guojin Qin

Academic Editor

PLOS ONE

Additional Editor Comments (optional):

Although the authors have addressed the concerns of reviewers, the quality of the figures should be improved, especially the clarity.
---

## [Editor Report · Acceptance letter]

PONE-D-25-29855R2

PLOS ONE

Dear Dr. Tardini Paulino,

I'm pleased to inform you that your manuscript has been deemed suitable for publication in PLOS ONE. Congratulations! Your manuscript is now being handed over to our production team.

Kind regards,

on behalf of

Dr. Guojin Qin

Academic Editor

PLOS ONE